# Betulinic Acid Attenuates Oxidative Stress in the Thymus Induced by Acute Exposure to T-2 Toxin via Regulation of the MAPK/Nrf2 Signaling Pathway

**DOI:** 10.3390/toxins12090540

**Published:** 2020-08-22

**Authors:** Lijuan Zhu, Xianglian Yi, Chaoyang Ma, Chenxi Luo, Li Kong, Xing Lin, Xinyu Gao, Zhihang Yuan, Lixin Wen, Rongfang Li, Jing Wu, Jine Yi

**Affiliations:** 1Colleges of Veterinary Medicine, Hunan Agricultural University, Changsha 410128, China; zhulijuan@stu.hunau.edu.cn (L.Z.); yixianglianhj@hotmail.com (X.Y.); mcyhndf@stu.hunau.edu.cn (C.M.); luochenxione@stu.hunau.edu.cn (C.L.); kongli@stu.hunau.edu.cn (L.K.); linxing@stu.hunau.edu.cn (X.L.); Gaoxinyu@stu.hunau.edu.cn (X.G.); zhyuan2016@hunau.edu.cn (Z.Y.); wenlixinedu@hotmail.com (L.W.); lirongfang@hunau.edu.cn (R.L.); 2Hunan Engineering Research Center of Livestock and Poultry Health Care, Changsha 410128, China; 3Hunan Co-Innovation Center of Animal Production Safety, Changsha 410128, China

**Keywords:** T-2 toxin, thymus, betulinic acid, MAPK/Nrf2, oxidative stress

## Abstract

T-2 toxin, the most toxic of the trichothecenes, is widely found in grains and feeds, and its intake poses serious risks to the health of humans and animals. An important cytotoxicity mechanism of T-2 toxin is the production of excess free radicals, which in turn leads to oxidative stress. Betulinic acid (BA) has many biological activities, including antioxidant activity, which is a plant-derived pentacyclic triterpenoid. The protective effects and mechanisms of BA in blocking oxidative stress caused by acute exposure to T-2 toxin in the thymus of mice was studied. BA pretreatment reduced ROS production, decreased the MDA content, and increased the content of IgG in serum and the levels of SOD and GSH in the thymus. BA pretreatment also reduced the degree of congestion observed in histopathological tissue sections of the thymus induced by T-2 toxin. Besides, BA downregulated the phosphorylation of the p38, JNK, and ERK proteins, while it upregulated the expression of the Nrf2 and HO-1 proteins in thymus tissues. The results indicated that BA could protect the thymus against the oxidative damage challenged by T-2 toxin by activating Nrf2 and suppressing the MAPK signaling pathway.

## 1. Introduction

T-2 toxin is produced by a species of the fungal genus *Fusarium* with highly toxicity. T-2 toxin was first isolated by Bamburg in 1968 [1]. In 1973, the World Health Organization and the United Nations Food and Agriculture Organization (FAO) listed this toxin, which is similar to aflatoxins, as the most dangerous source of food contamination in nature. T-2 toxin is widely distributed in nature and can easily contaminate animal feed and the environment. According to the abovementioned report, contamination with T-2 toxin has caused problems in many countries [2]. For example, T-2 toxin and the similar HT-2-toxin were detected in 29% and 16%, respectively, of 188 cereal and bedding straw samples collected in 2011 to 2012 from Swedish pig farms [3]. Additionally, a survey of mycotoxin contamination in 420 feedstuff samples collected from Shandong province in China found that the incidence of fumonisin B1, zearalenone, and T-2 toxin was 96.1%, 85.2%, and 79.5%, respectively [4]. It is reported that T-2 toxin is widely found in cereals, agricultural products, and animal feeds [5], and the ingestion by people or animals of grains or feed contaminated with T-2 toxin can cause various acute and chronic poisoning symptoms. Symptoms of acute poisoning include vomiting, hemorrhagic dysfunction, and cardiovascular dysfunction (such as endotoxin shock) [6], while chronic poisoning symptoms include loss of appetite, weight loss, and oral and esophageal lesions, among others [7]. According to some studies, T-2 toxin has many toxic effects, include immunotoxicity [8], reproductive toxicity [9], hematotoxicity [10], hepatotoxicity [2], neurotoxicity [11], cardiotoxicity [12], and bone system damage [13] (Table 1).

T-2 toxin inhibits the synthesis of eukaryotic proteins by oxidative stress. This kind of oxidative stress is caused by excessive production of reactive oxygen species (ROS) [14]. T-2 toxin mainly damages tissues and organs in which vigorous cell division occurs, such as lymphoid, hematopoietic, and gastrointestinal tissues [15]. Lymphatic organs are an important part of the body’s immune system and play roles in preventing many diseases. T-2 toxin can reduce the splenic follicles and thymus cortex lymphocytes number and induce apoptosis in the lymphoid organs of mice [16]. Further, Nagata [17] found that thymocytes are more sensitive to T-2 toxin-induced apoptosis than lymphocytes in Peyer’s patches and mesenteric lymph nodes. Therefore, in this study, the thymus of mice was selected to establish the model of T-2 toxin-induced tissue damage.

Pentacyclic triterpenoids are a class of triterpenoids that are widely found in natural plants, and have been investigated for their pharmacological activities. Betulinic acid (BA) is a triterpenoid made from a lupane-structured pentacyclic triterpene [18], which is mainly extracted from the bark of birch trees. According to some studies, BA possesses many biological activities, including immunomodulatory, antioxidation, anti-tumor, anti-diabetic, anti-tuberculosis, anti-viral, and anti-inflammatory activities, as well as some anti-microbial properties [19,20,21,22,23]. BA was previously found to be able to ameliorate psoriasis-like murine skin inflammation by decreasing the number of γδ T cells and IL-17A-expressing CD4+ in psoriatic mice, and inhibiting the proliferation of T cell and IL-17A production by CD4+ T-Cells in vitro [20]. Furthermore, BA can increase immunomodulatory activity by inducing the activation of macrophages [24]. Our previous study reported that BA enhanced mouse macrophage activity, humoral immunity, and cellular immunity. [25]. BA also showed protective activities against alcohol-induced liver damage and dexamethasone-induced spleen and thymus oxidative damage, and these protective effects were related to the antioxidant capacity of BA [23,26]. Thus, it is worth investigating whether BA can regulate the oxidative damage in the thymus in mice caused by T-2 toxin through its antioxidant ability. In this study, the protective effect and possible mechanisms of BA in reducing acute exposure to T-2 toxin-induced oxidative damage in immune organs were explored. This could provide a regulatory target for relieving oxidative stress caused by T-2 toxin and explore the feasibility of BA’s application in livestock and poultry industry.

## 2. Results

### 2.1. Effect of BA on Immunoglobulin G (IgG) and IgM Induced by T-2 Toxin in Serum

Serum immunoglobulin level is an important indicator reflecting the state of the immune system. In this study, T-2 toxin had no significant effects on IgG and IgM compared with the control group (*p* > 0.05). However, BA pretreatment significantly increased the content of IgG at the dosage of 0.5 mg/kg (*p* < 0.01), while it had no effects on IgM, compared with the T-2 toxin group (Figure 1). 

### 2.2. BA Decreased T-2 Toxin-Induced Thymus ROS and MDA Generation

It was investigated whether the therapeutic administration of BA could protect against the rise in ROS levels caused by T-2 toxin via ROS scavenging by BA (Figure 2). The size of the red-staining area increased significantly, and the ROS level was enhanced in the thymus of mice treated with T-2 toxin, compared with the control group (*p* < 0.01). However, BA pretreatment significantly decreased (*p* < 0.01) the level of ROS, compared with that in the T-2 toxin group in a dose-dependent manner. Besides, T-2 toxin could significantly increase (*p* < 0.05) the content of malondialdehyde (MDA), while this increasing of the content of MDA in the thymus was inhibited (*p* < 0.05) at a dosage of 1.0 mg/kg BA.

### 2.3. BA Increased T-2 Toxin-Induced Thymus Antioxidative Capacity

The effects of BA on T-2 toxin-induced thymus antioxidative capacity are presented in Figure 3. T-2 toxin significantly decreased (*p* < 0.01) the levels of total antioxidative capacity (T-AOC), superoxide dismutase (SOD), and glutathione (GSH), compared with the control group. However, the levels of SOD and GSH were enhanced (*p* < 0.01) at all dosages of BA, while there was no significant effect (*p* > 0.05) on the T-AOC level of the thymus.

### 2.4. Effect of BA on Morphological Change of the Thymus

The results of hematoxylin-eosin (H&E) staining of thymus tissues in different treatments are shown in Figure 4A–E. The cells in the thymus were arranged regularly, clearly and without apoptosis. (Figure 4A). After T-2 toxin administration (Figure 4B), some of the cells in the thymus were necrotic and apoptotic, and the nucleus stained deeply and fragmented or dissolved, and the cells were arranged loosely. Simultaneously, vascular congestion could be seen in the thymus after T-2 toxin administration (Figure 4B). Necrosis and apoptosis in the cells of the thymus were still observed after BA pretreatment, but the thymic congestion caused by T-2 toxin was improved by BA (Figure 4C–E).

### 2.5. BA Suppressed MAPK Signaling Pathway Induced by T-2 Toxin in the Thymus

The phosphorylation and expression of MAPK signaling pathway-related proteins, such as *c-jun* N-terminal kinase (JNK), phosphorylated JNK (p-JNK), p38 mitogen-activated protein kinase (p38), p-p38, extracellular signal-regulated kinases (ERK), and p-ERK were detected in the current study. In Figure 5, a significant increase (*p* < 0.01) in the phosphorylated to unphosphorylated ratios of such proteins (p-p38/p38, p-ERK/ERK, and p-JNK/JNK) was recorded in T-2 toxin-induced mice. However, BA significantly decreased (*p* < 0.01) the extent to which the p-p38/p38 ratio was increased by T-2 toxin in a dose-dependent manner. Further, BA significantly reduced the p-ERK/ERK and p-JNK/JNK ratios compared with those in the T-2 toxin group at dosages of 0.25 and 1.0 mg/kg (*p* < 0.01).

### 2.6. BA Activated Nuclear Factor Erythroid 2 [NF-E2]-Related Factor 2 (Nrf2)—Heme Oxygenase-1 (HO-1) Signaling Pathway Induced by T-2 Toxin in the Thymus

The Nrf2-HO-1 signaling pathway is closely related to oxidative stress. Thus, the expression level of proteins in the Nrf2-HO-1 signaling pathway were detected. The expression of Kelch-like erythroid cell-derived protein with CNC homology [ECH]-associated protein 1 (Keap1) protein was significantly increased (*p* < 0.01), while the expression levels of Nrf2 and HO-1 proteins were decreased (*p* < 0.01 and *p* < 0.05, respectively) in the T-2 toxin group, when compared with the control group. However, pretreatment with BA significantly increased the expression of HO-1 protein at all dosages (*p* < 0.01 and *p* < 0.05, respectively), and also increased the expression of Nrf2 protein at a high BA dosage (*p* < 0.01). Notably, BA significantly increased (*p* < 0.05) the expression of Keap1 protein compared with that in the T-2 toxin group (Figure 6).

## 3. Discussion

In recent years, contamination by mycotoxins has caused enormous economic losses to agricultural and livestock production and great harm to human health. According to the FAO, approximately 25% of agricultural crops worldwide are contaminated by mycotoxins to varying degrees each year, and almost 4.5–5 billion people are at risk of chronic exposure to mycotoxins due to serious pollution issues [27]. Trichothecenes are one of the most important types of mycotoxins in agriculture that also pose potential hazards to global health [28]. T-2 toxin is a class A trichothecene and is the most acutely toxic among these kinds of toxins [29]. This study showed that a single dose of T-2 toxin at dosage levels of 0.5 mg/kg and 2.0 mg/kg only influenced the urinary metabonomes, while 4.0 mg/kg T-2 toxin induced metabolic alterations in urine and multiple organs. T-2 toxin also causes oxidative stress and disturbance in energy metabolism and gut microbiome [30]. Besides, research has shown that intraperitoneal injection of T-2 toxin also causes oxidative stress [31]. Our previous study found that intraperitoneal injection of 4 mg/kg T-2 toxin could cause oxidative damage in testicular tissue [32]. Based on these studies, an intraperitoneal injection of 4 mg/kg T-2 toxin was used to establish an oxidative damage model of the thymus in this study.

T-2 toxin has cytotoxic and immunosuppressive effects in DNA and RNA synthesis [33]. IgG and IgM are important immunoglobulins for immune response, accounting for about 75~80% and 5~10% of total antibodies, respectively. Several studies have reported that T-2 toxin suppressed induction of IgG and IgM [34,35]. BA showed an immunomodulatory property by producing pro-inflammatory cytokines and activation of macrophages [24]. In the present study, T-2 toxin had no effects on the contents of IgG and IgM at a dosage of 4 mg/kg. This may be because 4 mg/kg of T-2 toxin is not enough to cause the change in IgG and IgM contents. However, the BA medium-dose group was able to significantly increase serum IgG content, while it had no effect on IgM of the thymus in mice. This was probably because IgM was the first antibody secreted in immune response. Once infected, it is produced quickly. After a period of time, the amount of IgM antibody gradually decreases and disappears.

T-2 toxin exerts cytotoxicity by causing oxidative stress through the overproduction of ROS [36]. Under normal physiological conditions, ROS production is regulated by enzymatic antioxidants, including GSH, catalase (CAT), and SOD, and other non-enzymatic antioxidants. However, when the antioxidant defense capabilities are lower than the production of ROS of biological tissues, oxidative stress may have deleterious effects on their function and structural integrity [37]. T-AOC is also an important indicator of the activity of the antioxidant defense system [23]. SOD is a major component of the antioxidant system, while GSH maintains cellular redox balance [38,39]. When GSH and SOD levels are reduced, this leads to excessive utilization of superoxide and hydrogen peroxide, which leads to lipid peroxidation by hydroxyl radicals and increases the cellular content of MDA [40,41]. The study showed that T-2 toxin suppressed the viability of granulosa cells and caused apoptosis, which was accompanied by the accumulation of ROS and MDA and decreasing levels of CAT, SOD, and GSH-Px in these cells [42]. Furthermore, the potential mechanism of the apoptosis and DNA damage caused by T-2 toxin is oxidative stress, which may be mediated in part through the accumulation of ROS, depletion of GSH levels, and increased lipid peroxidation. As a result, human cervical cancer cells exhibit apoptotic morphology, such as condensed chromatin and nuclear fragmentation, processes which promote the expression of pro-apoptotic proteins and inhibit the expression of anti-apoptotic proteins [43]. Consistent with these results, T-2 toxin increased ROS content; decreased the levels of GSH, SOD, and T-AOC; and increased the content of MDA in the thymus in the current study. After T-2 toxin administration, many cells in the thymus also became necrotic and apoptotic, the nucleus was stained deeply and fragmented or dissolved, and the cells of the thymus became loosely arranged. This indicates that an induced model of oxidative stress was established in an immune organ of mice using T-2 toxin.

BA is a natural pentacyclic triterpenoid extracted from birch trees, for which many biological activities have been demonstrated in previous studies, including antioxidative activity. For example, BA pretreatment can prevent kidney damage by recovering the imbalance of inflammatory mediators, oxidants, and antioxidants induced by cecal ligation puncture [44]. Our previous studies also demonstrated that BA could improve the antioxidant capacity of tissues by decreasing the content of MDA and increasing the levels of GSH-Px and SOD in thymocytes and splenocytes, respectively, and also that it could have protective effects against the oxidative stress induced by dexamethasone [45,46]. In this study, BA pretreatment significantly decreased the production of ROS. Further, BA significantly ameliorated the decreases in SOD and GSH levels and the increase in MDA content in the thymus caused by T-2 toxin. This indicated that BA can function as an antioxidant by scavenging the majority of free radicals in cells and attenuate oxidative stress by increasing cellular antioxidant capacity and reducing the incidence of lipid peroxidation. It is noteworthy that BA had no significant destructive effect on tissue structure or on the inflammatory response, but rather was able to ameliorate the thymic congestion caused by T-2 toxin. This shows that BA could protect lymphatic organs by reducing thymic congestion without harmful side-effects.

Oxidative stress causes cell apoptosis, which mostly occurs via a cascade of apoptosis-related signaling pathways, including the MAPK pathway [23,47]. ERK, JNK, and p38 are the most frequently studied among the MAPK signaling pathway. Many researches have proved that oxidative stress conditions can activate the p38, JNK, and ERK signaling pathways, leading them to induce apoptosis [48,49]. T-2 toxin has toxic effects by causing oxidative stress, which subsequently activates the MAPK pathway, and ultimately induces apoptosis in the liver of pregnant mice [50]. Further, the *c-jun* gene, an early gene which is regulated by JNK, may play an important role in T-2 toxin-induced apoptosis [50]. Previous research found that oxidative stress caused by T-2 toxin induces skin inflammation and cutaneous damage by activating p38, MAPK, and epidermal cell apoptosis pathways, leading to changes in skin histology [51]. In the current study, T-2 toxin significantly increased the ratios of p-ERK/ERK, p-p38/p38, and p-JNK/JNK. This suggests that the T-2 toxin activated the MAPK signaling pathway by increasing the phosphorylation of p38, JNK, and ERK, which ultimately led to oxidative damage to the thymus. Eliminating a large number of ROS-induced apoptosis signaling molecules through the use of antioxidants is a key method to inhibit oxidative stress-induced cell damage. BA showed protective effects from ischemia/reperfusion injury (I/RI) in H9c2 cells by suppression cell apoptosis and oxidative stress, and its potential cardio-protective mechanisms were related to the JNK, p38, and Nrf2/HO-1 pathways [52]. BA can also downregulate ERK expression, and was shown to have antioxidative activity in many previous studies [53]. In our previous study, it was found that BA inhibited the expression of p38 and JNK, thereby inhibiting the mitochondrial apoptosis pathway, and ultimately reducing the oxidative damage to and apoptosis of lymphocytes induced by dexamethasone [23]. In the current study, BA protected against T-2 toxin-induced oxidative damage to the thymus by inhibiting the expression and phosphorylation of the p38, JNK, and ERK proteins involved in the MAPK signaling pathway in thymus tissues. These results demonstrated that BA could alleviate oxidative stress in the body by inhibiting the expression of p38, JNK, and ERK proteins and their phosphorylation, and therefore it could have protective effects against the oxidative damage challenged by T-2 toxin.

The Nrf2 signaling pathway is a classic antioxidative stress pathway; its activation is involved in cellular responses to oxidative stress and results in up-regulation of the antioxidant defense mechanisms. Under normal physiological conditions, Nrf2 is kept in the cytoplasm by Keap1 through the formation of Keap1-Nrf2 complexes. When oxidative stress occurs, the Keap1-Nrf2 complex is decoupled, and Nrf2 is released from the cytoplasm into the nucleus [54]. After Nrf2 enters the nucleus, it links to antioxidant response elements that mediate the antioxidant genes’ expression, and participates in the transcription of downstream phase II metabolic enzyme genes, including HO-1, resulting in antioxidative activity [55]. HO-1 is the rate-limiting enzyme for heme metabolism and belongs to the heat shock protein family, and also has obvious antioxidant effects [56]. In the present study, treatment with T-2 toxin significantly increased the expression of Keap1 protein but reduced the expression of Nrf2 and HO-1 proteins. Similar results have been obtained in previous studies [33], and suggested that T-2 toxin-induced immunotoxicity in the thymus involves oxidative stress induced by promoting the expression of Keap1, which thereby leads to Nrf2 being trapped in the cytoplasm and reduces the downstream expression of the antioxidant protein HO-1. Antioxidants that remove oxidative factors and activate antioxidant signaling pathways are often used as a method to inhibit oxidative damage. BA was previously reported to enhance the activation of the Nrf2/HO-1 pathway and to have protective effects against I/RI in H9c2 cells [52]. Besides, another study also showed that N-acetylcysteine, a commonly used antioxidant, protected against ischemia renal injury by increasing Nrf2 and downstream HO-1 expression, and decreasing cleaved caspase 3 and p53 expression, which decreased renal epithelial tubular cell apoptosis [57]. In the current study, BA pretreatment significantly increased Nrf2 and HO-1 protein expressions. However, pretreatment with BA promoted the expression of Keap1 relative to that in the T-2 toxin group. Although Keap1 is the most important regulator of Nrf2, Nrf2 can also be regulated by protein kinase C, ERK, p38, and JNK [58,59]. Thus, in the current study, BA increased the expression of nuclear Nrf2 through other pathways than that regulated by Keap1. These results indicated that BA had a good antioxidant effect, which was achieved by promoting the expression of nuclear Nrf2 and activating the antioxidant enzyme HO-1. However, it is still unclear how BA promoted the expression of Keap1 protein, so this needs to be examined with further experiments.

Taken together, our results illustrated that BA effectively relieved T-2 toxin-induced oxidative damage in immune organs by scavenging ROS, reducing lipid peroxidation, and enhancing the antioxidant capacity in the thymus. Furthermore, by inhibiting the MAPK signaling pathway and activating the Nrf2 antioxidative stress signaling pathway, BA enhanced the body’s antioxidant capacity and protected the immune organs from T-2 toxin-caused oxidative damage. This indicated that BA could protect the thymus against the oxidative damage challenged by T-2 toxin through activation of Nrf2 and suppression of the MAPK signaling pathway.

## 4. Methods

### 4.1. Reagents

T-2 toxin was obtained from Pribolab Pte. Ltd. (Singapore). BA was obtained from Sigma (St Louis, MO, USA). BCA^TM^, MDA, GSH, SOD, and T-AOC assay kits were obtained from Nanjing Jiancheng Biotech (Nanjing, Jiangsu, China). IgG (E-AB-1033) and IgM (E-EL-M3036) enzyme-linked immunosorbent assay (ELISA) kits were provided by Elabscience Biotechnology Co., Ltd. (Wuhan, Hubei, China). Beyotime Institute of Biotechnology (Shanghai, China) provided the radio immunoprecipitation assay (RIPA) buffer. Dihydroethidium (DHE) dye solution and H&E staining materials were purchased from Wuhan Google Biotechnology Co., Ltd. (Wuhan, Hubei, China). Jiangsu Keygen Biotech Corp., Ltd. (Jiangsu, China) provided the enhanced chemiluminescence (ECL) reagent. Antibodies, such as those against β-actin (3700S), histone H3 (4499S), JNK (9252S), p-JNK (9251S), ERK (9102S), p-ERK (9101S), p38 (9212S), p-p38 (9211S), Nrf2 (12721S), HO-1 (82206S), and Keap1 (8047S) were provided by Cell Signaling Technology, Inc. (Danvers, MA, USA).

### 4.2. Experimental Design

Fifty male Kunming mice (4–5 w) were provided by Hunan Silaike Jingda Laboratory Animal Co., Ltd. (Changsha, Hunan, China). The duration of treatment and the doses of T-2 toxin and BA were chosen from previous studies and preliminary experiments [23,32]. The mice were placed individually in a room with controlled humidity (50–70%) and temperature (22–25 °C). After 1 week of pre-feeding, the mice were randomly divided into the control group, the T-2 toxin group, the low dose of BA group (0.25 mg/kg), the medium dose of BA group (0.5 mg/kg), and the high dose of BA group (1.0 mg/kg). Due to the poor solubility of BA in water, BA was suspended in 1% starch jelly in this experiment. For the first 14 days, the control group, the T-2 toxin group, and the low, medium, and high dose of BA groups were sequentially administered 0, 0, 0.25, 0.5, and 1.0 mg of BA per kg of body weight, respectively. The control group received an intraperitoneal injection with a 1:12.5 mixture of ethanol and phosphate-buffered saline (PBS), while the other 4 groups were treated with T-2 toxin in the same way with a dosage of 4 mg/kg (T-2 toxin was dissolved in ethanol to a concentration of 5 mg/mL, and then diluted in PBS with the ratio 1:12.5) to create a model of induced oxidative damage after administration of BA. Fifteen hours later, the blood samples were collected by venous puncture after anesthetization with diethyl ether and the mice were dissected. The thymus was quickly excised and placed in sterile saline solution to rinse off the blood. One part of the thymus was homogenized to form a 10% homogeneous sample mixed with a cold physiological saline. After centrifuging (3000 rpm for 15 min at 4 °C), the supernatant was then aspirated and deposited at −80 °C for analysis. A part of the thymus was placed in a 4% paraformaldehyde (PFA) solution for morphological observation. One part of the thymus was made into frozen slices for ROS detection. The remaining of the thymus was deposited at −80 °C for western blot analysis.

All experimental procedures were conducted as per the Animal Care and Use Guidelines of China and authorized by the Animal Care Committee. The project approval code is 43321820 and approval date is 1 March 2018.

### 4.3. Evaluation of IgG and IgM

The blood samples were put in a refrigerator at 4 °C for 12 h, and then the supernatant was collected after the samples were centrifuged at 3000 rpm for 10 min. IgG and IgM were detected by ELISA kits in serum, while strictly following the directions in the instruction manual provided with each respective kit.

### 4.4. Intracellular ROS Measurement

The production of ROS was measured in the frozen thymus sections using the oxidative fluorescent dye, DHE. The dye was applied to the thymus sections, and the slides were then incubated for 30 min at 37 °C in a light-protected humidified chamber. The fluorescence was then measured to be between an emission wavelength of 610 nm and excitation wavelength of 535 nm with a fluorescence microscope system (Olympus, Tokyo, Japan). Because DHE could freely penetrate the living cell membrane into the cell, it was oxidized by intracellular ROS to form ethidium oxide. The ethidium oxide was then able to incorporate into chromosomal DNA to produce red fluorescence. Thus, according to the production of red fluorescence, the levels of intracellular ROS were quantified by Image-Pro Plus software [60].

### 4.5. Evaluation of Antioxidative Capacity

The supernatant produced after processing thymus samples as described above was collected and analyzed to detect the levels of MDA, SOD, T-AOC, and GSH in the thymus, while strictly following the directions in the instruction manual provided with each respective kit. The content of MDA was measured with the thiobarbituric acid method, while the content of GSH was measured with the dithiodinitrobenzoic acid method. The T-AOC was measured by chemical colorimetry method, and the activity of SOD was measured by the xanthine oxidase method.

### 4.6. Histopathology of the Thymus

The PFA-fixed thymus samples were embedded in paraffin and cut into sections. The paraffin sections of the thymus were then stained with H&E using standard methods, and then finally dehydrated with a neutral gum seal. The paraffin sections were observed and analyzed under light microscopy (Nikon Eclipse Ci, Tokyo, Japan).

### 4.7. Western Blot Analysis

Western blotting is based on the methods commonly used in our laboratory [23]. Samples of the thymus were ground up while adding an RIPA lysis buffer containing phenylmethanesulfonyl fluoride (PMSF). The thymus lysates were collected after centrifugation, and then the protein concentrations were determined. The thymus samples were loaded into sodium dodecyl sulfate polyacrylamide gel electrophoresis (SDS-PAGE) gels for electrophoresis. The protein band from SDS-PAGE was transferred onto polyvinyldifluoride membranes, which were then blocked for 1 h. The membranes were incubated overnight with primary antibodies, such as β-actin, p38, ERK, JNK, p-p38, p-ERK, p-JNK, Keap1, and HO-1. The membranes were incubated with secondary antibodies and washed thrice with TBST. Each protein band was then quantified by densitometry and analyzed using Alpha software (Alpha Innotech, San Leandro, CA, USA).

For analysis of the nuclear protein Nrf2 and histone H3, thymus homogenate was prepared using a PMSF-containing cytoplasmic protein extraction reagent. After centrifugation, the supernatant was aspirated and abandoned. The nuclear pellets were resuspended in the buffer solution, and every 1–2 min vortexed vigorously for 15–30 s for 30 min on ice. The supernatant was removed after centrifugation, and then the concentrations of these proteins in it were determined. These samples were also used to perform western blot analyses, as described above.

### 4.8. Statistical Analysis

SPSS software version 17.0 was used for the statistical analysis. The significance of differences among treatments was determined by performing analysis of variance (ANOVA), followed by Tukey’s least significant difference (LSD) test. The results are expressed herein as the mean ± standard error of mean (SEM). Differences were considered as statistically significant at *p* < 0.05.

## Figures and Tables

**Figure 1 toxins-12-00540-f001:**
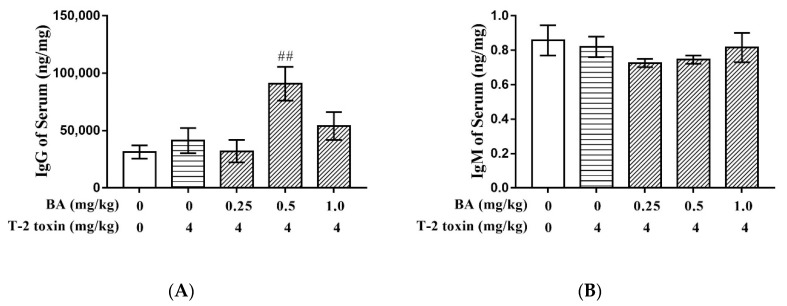
Effect of BA on IgG (**A**) and IgM (**B**) induced by T-2 toxin in serum: Values are presented as the mean ± standard error of mean (SEM) content in each treatment. ##: *p* < 0.01 compared to the T-2 toxin group.

**Figure 2 toxins-12-00540-f002:**
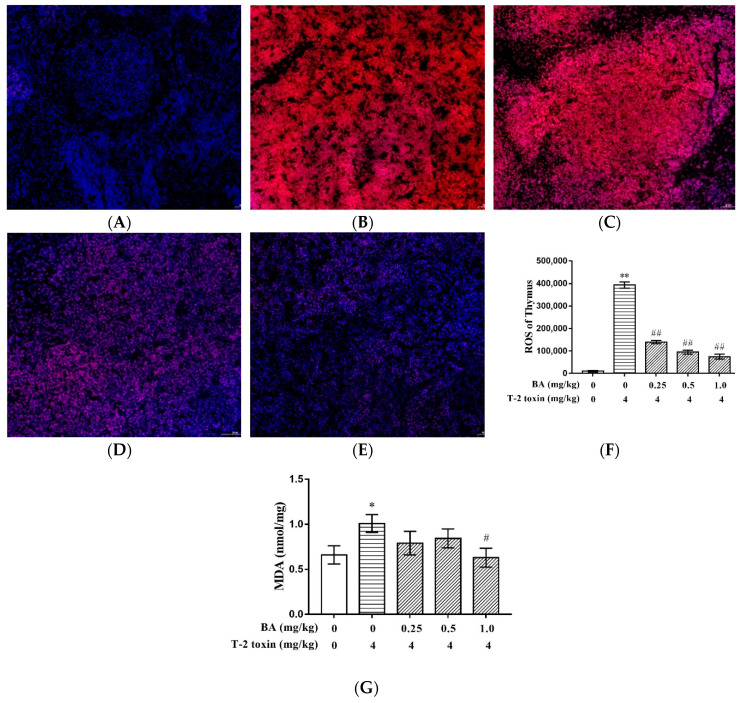
BA decreased T-2 toxin-induced thymus ROS and MDA generation: Fluorescent micrographs are shown for the control (**A**), T-2 toxin (**B**), low dose of BA (**C**), medium dose of BA (**D**), and high dose of BA (**E**) groups. ROS levels were determined under a fluorescence microscope and stained by DHE; scar bar: 50 μm. The ROS level (**F**) and the MDA (**G**) content were decreased after pretreatment with BA in the T-2 toxin-treated thymus. Values are presented as the mean ± SEM in each treatment. *: *p* < 0.05 compared to the control group; **: *p* < 0.01 compared to the control group; #: *p* < 0.05 compared to the T-2 toxin group; ##: *p* < 0.01 compared to the T-2 toxin group.

**Figure 3 toxins-12-00540-f003:**
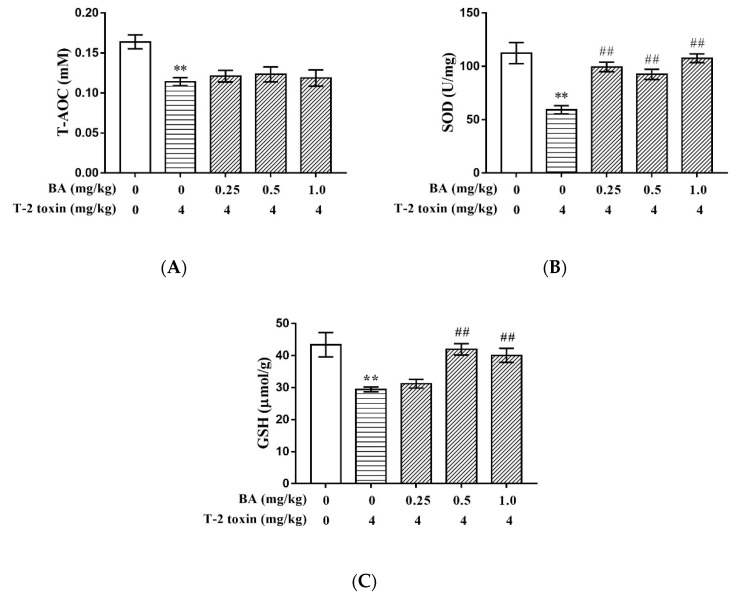
Effect of BA on the levels of T-AOC (**A**), SOD (**B**), and GSH (**C**) in the thymus of mice induced by T-2 toxin. Values are presented as the mean ± SEM content in each treatment. **: *p* < 0.01 compared to the control group; ##: *p* < 0.01 compared to the T-2 toxin group.

**Figure 4 toxins-12-00540-f004:**
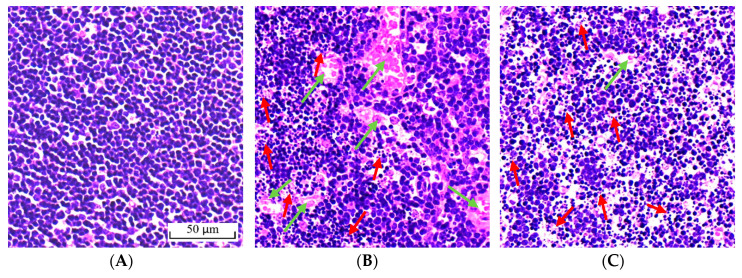
Effect of BA on the histopathological characteristics of the thymus in mice induced by T-2 toxin. No apoptosis was observed in the control group (**A**). Apoptosis and nuclear fragmentation were observed in mice in the T-2 toxin group (**B**). Thymic congestion was alleviated by the administration of BA at dosages of 0.25 mg/kg (**C**), 0.5 mg/kg (**D**), and 1.0 mg/kg (**E**). Red arrows: necrotic and apoptotic; green arrow: congestion; scale bar: 50 μm.

**Figure 5 toxins-12-00540-f005:**
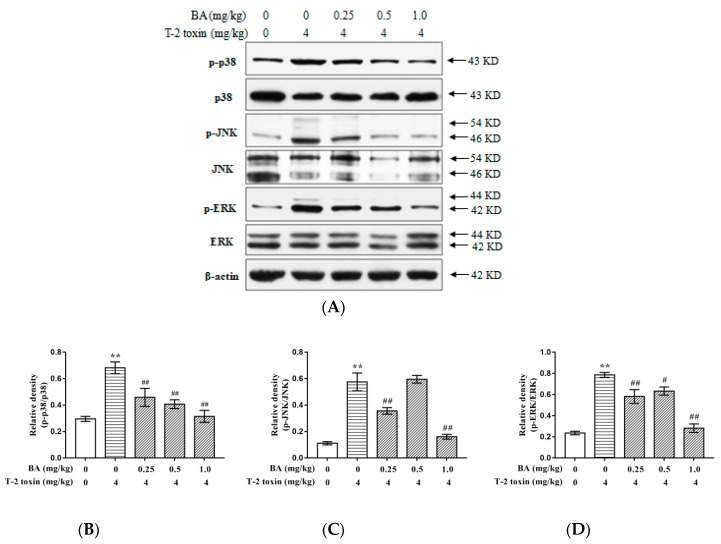
BA suppressed MAPK signaling pathway induced by T-2 toxin in the thymus: The protein phosphorylation and expression levels of p38, JNK, and ERK (**A**) in the thymus were measured using Western blotting. Western blotting of analysis of BA on expression of p38 (**B**), JNK (**C**), and ERK (**D**). The thymus lysis samples were subjected to western blot analysis with anti-p-p38, anti-p38, anti-p-JNK, anti-JNK, anti-p-ERK, and anti-ERK antibodies. The data were expressed as the ratios of p-p38/p38, p-JNK/JNK, and p-ERK/ERK. Values are presented as the mean ± SEM value in each treatment. **: *p* < 0.01 compared to the control group; #: *p* < 0.05 compared to the T-2 toxin group; ##: *p* < 0.01 compared to the T-2 toxin group.

**Figure 6 toxins-12-00540-f006:**
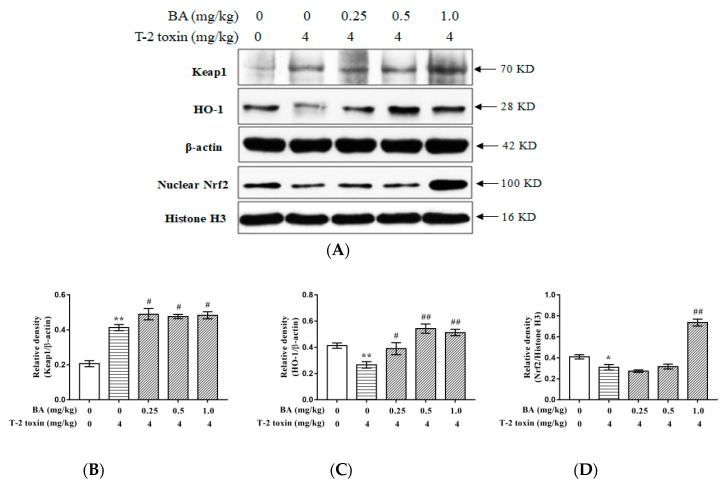
BA activated Nrf2-HO-1 signaling pathway induced by T-2 toxin in the thymus: The protein expressions of the Keap1, HO-1, and Nrf2 (**A**) in the thymus were detected using western blotting. Western blotting of analysis of BA on Keap1 (**B**), HO-1 (**C**), and Nrf2 (**D**). The thymus lysis samples were used to western blot analysis incubated with anti-Keap1, anti-HO-1, and anti-Nrf2 antibodies. The data were expressed as the ratios of Keap1/β-actin, HO-1/β-actin, and Nrf2/Histone H3. Values are presented as the mean ± SEM value for each treatment. *: *p* < 0.05 compared to the control group; **: *p* < 0.01 compared to the control group; #: *p* < 0.05 compared to the T-2 toxin group; ##: *p* < 0.01 compared to the T-2 toxin group.

**Table 1 toxins-12-00540-t001:** Toxic effects of T-2 toxin.

Toxic Effects	Results	References
Immunotoxicity	T-2 toxin determined sustained (48 h) immunosuppression on human lymphoid cell lines of T or B lineage cell lines.	[8]
Reproductive toxicity	T-2 toxin impaired male fertility by disrupting the hypothalamic-pituitary-testis axis and declining testicular function in mice.	[9]
Hematotoxicity	Hypovolemia with polycythemia resulting from plasma leakage and internal bleeding accounts for acute lethal T-2 toxin.	[10]
Hepatotoxicity	DNA methylation regulated the RASSF4 expression under T-2 toxin, along with the activation of its downstream pathways, resulting in liver apoptosis.	[2]
Neurotoxicity	T-2 toxin induced autophagy in the brain and apoptosis in the pituitary in rat.	[11]
Cardiotoxicity	T-2 toxin significantly increased the intensity of myocardial degeneration and haemorrhages, distribution of glycogen granules in the endo- and perimysium, a total number of mast cells and the degree of their degranulation.	[12]
Bone system damage	T-2 toxin can cause damage to articular cartilage and weight loss in rats, which may be related to the Ihh-PTHrP pathway.	[13]

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
