# Peer review of "Betulinic Acid Attenuates Oxidative Stress in the Thymus Induced by Acute Exposure to T-2 Toxin via Regulation of the MAPK/Nrf2 Signaling Pathway"

_toxins, 2020, doi:10.3390/toxins12090540_

Round 1
Reviewer 1 Report
The manuscript „Betulinic acid attenuates oxidative stress in the thymus induced by T-2 toxin via regulation of the MAPK/Nrf2 signaling pathway“ addresses an important aspect of the function of betulinic acid in interfering with the toxic functions of T-2 toxin - especially in immune organs.
The authors used for their study Kunming-mice, the doses of T-2 and betulinic acic are appropriate as well as the experimental design.
In the Results part the evaluation of the immunoglobulin-levels is interesting, however, why is there a decline in the IgG level at 1.0 mg/kg BW betulinic acid in comparison to 0.5 mg?
The analysis of the reactive oxygen species (ROS) levels is not clearly described in the Methods part. There is no reference given, and there is no description of the criteria of the morphometry and further microscopical analysis of the morphological sections.
As the cellular methods for the thymic antioxidative capacity tests is not given in detail, the results given in Fig. 3 can not be commented.
The histopathological study (Fig. 4) is far beyond any microscopical standad. The tissue sections are unclear and much to thick. So the pictures A to E are of reduced quality, apoptotic cells may be in the figures - but the red arrows point in many locations to anywhere. Whether the green arrows indicate tissue congestion can not be answered. What „congestion“ may be is not given in detail. No further criteria how the morphometry is done - and what statistics are the basis?
The Western Blots in Fig. 5 and 6 are interesting, why is there in the upper row, Fig. 6, the Keap1-band at 1.0 betulinic acid so staggered?
The Discussion part is very extensive, but often not dedicated to the findings of the study.
In total, especially because of the poor morphology/morphometry and ROS analysis from thymic samples in the context of less well described methods , although the scientific question is of interest.
Author Response
Responses to Reviewer 1 Comments
Point 1: In the Results part the evaluation of the immunoglobulin-levels is interesting, however, why is there a decline in the IgG level at 1.0 mg/kg BW betulinic acid in comparison to 0.5 mg?
Response 1: We are very grateful to the reviewer for the comments. In general, the effect of the medicines, within a certain dose range, increase as the increasing concentration of the medicines. However, it will decrease efficacy, and ever produce adverse effect when the dosages continue to rise and reach the tolerable upper intake level. So, not the higher the dose, the better the effect. For example, manganese is an essential nutrient which can cause adverse effects if ingested too excess or in insufficient amounts, leading to a U-shaped exposure-response relationship. (Milton, B.; Krewski, D.; Mattison, D.R.; et al. Modeling U-shaped dose-response curves for manganese using categorical regression. Neurotoxicology 2017, 58, 217-225). In our study, the medium dosage of BA (0.5 mg/kg) pretreatment had the most significant effect on the content of serum IgG. It means, BA intake couldn’t play the optimal protective role for oxidative stress if intake is either too high or too low. Similar to manganese, reduced efficacy can arise from exposure to extreme doses of BA. However, intake that is too low can also lead to low efficacy. Consequently, BA exhibits a U-shaped or inverted U-shaped exposure-response relationship: either too much or too little may result in low efficacy outcomes. Our result was to ensure sufficient BA intake to satisfy antioxidant properties against T-2 toxin-induced thymus damage, but not so large as to induce adverse health effects. Therefore, the concentration range of the potential thymus protection of BA in a dose-dependent manner warrants further investigation in the future.
Point 2: The analysis of the reactive oxygen species (ROS) levels is not clearly described in the Methods part. There is no reference given, and there is no description of the criteria of the morphometry and further microscopical analysis of the morphological sections.
Response 2: Thanks for the comments. The analysis method of ROS levels has been added in line 343-347.
Point 3: As the cellular methods for the thymic antioxidative capacity tests is not given in detail, the results given in Fig. 3 can not be commented.
Response 3: The detailed detection methods about the thymic antioxidative capacity have been added which could be found in “4.5 Evaluation of antioxidative capacity”.
Point 4: The histopathological study (Fig. 4) is far beyond any microscopical standard. The tissue sections are unclear and much to thick. So the pictures A to E are of reduced quality, apoptotic cells may be in the figures - but the red arrows point in many locations to anywhere. Whether the green arrows indicate tissue congestion can not be answered. What congestion“ may be is not given in detail. No further criteria how the morphometry is done - and what statistics are the basis?
Response 4: Thanks for your comments. We re-made the picture of H&E, the picture was enlarged, and the lesion area can be seen more clearly.
Point 5: The Western Blots in Fig. 5 and 6 are interesting, why is there in the upper row, Fig. 6, the Keap1-band at 1.0 betulinic acid so staggered?
Response 5: Thanks for the comments. Keap1-band at 1.0 mg/kg BA may not be perfect, but we have tried our best to do it.
Point 6: The Discussion part is very extensive, but often not dedicated to the findings of the study.
Response 6: Thanks for the comments. Change was made accordingly.
Reviewer 2 Report
In this work authors the protective effects and mechanisms of BA in blocking oxidative stress caused by T-2 toxin have been studied in the thymus of mice. The work is interesting with considerable addition to the knowledge in the field and surely will earn wide interest due to the importance. The references are up to date and comprehensive. The work is very nice and carefully discussed justifying publication in TOXINS. I suggest publication after considering the following minor remarks:
1) I am not sure under the description of the experiments, but I can assume that the experiments in serum performed in the presence of albumins. Did authors consider the possible adsorption of T-2 toxin on the surface of albumins ? If it could happen it would highly affect the results and would help to understand the hectic effect of BA. (Especially only in case of P-P38/P38, see Figure 5. B, can see the nice tendency.) This property do not affect the main conclusions, but maybe helps in the data evaluation. I suggest to authors to insert at least few sentences into the manuscript accordingly.
2) 4 mg/kg dosage of T-2 toxin looks very high. I suggest to insert few sentences to argue this dosage.
Minot tipos:
By my opinion, better to show the complete error bars on the figures instead of the only upper half.
Author Response
Responses to Reviewer 2 Comments
Point 1: I am not sure under the description of the experiments, but I can assume that the experiments in serum performed in the presence of albumins. Did authors consider the possible adsorption of T-2 toxin on the surface of albumins? If it could happen it would highly affect the results and would help to understand the hectic effect of BA. (Especially only in case of P-P38/P38, see Figure 5. B, can see the nice tendency.) This property do not affect the main conclusions, but maybe helps in the data evaluation. I suggest to authors to insert at least few sentences into the manuscript accordingly.
Response 1: Thanks for your suggestion very much, which provides some explanations for the trend of the indicators in the serum. However, we only found that T-2 toxin and bovine serum albumin can be cross-linked by succinic anhydride method. And we have not yet found relevant supporting studies for albumin to adsorb T-2 toxin. Of course, the serum index does not show a perfect trend, such as the BA medium-dose group can significantly increase the serum IgG content, while had no effect on IgM of thymus in mice. This was probably because IgM was the first antibody secreted in the immune response. Once infected, it is produced quickly. After a period of time, the amount of IgM antibody gradually decreases and disappears, which are also worth exploring.
Point 2: 4 mg/kg dosage of T-2 toxin looks very high. I suggest to insert few sentences to argue this dosage.
Response 2: Thanks for your comments. The dosage of T-2 toxin was determined on the basis of preliminary research and preliminary experiments. We have added relevant references in 4.2. Besides, there are many studies shown that the dose of T-2 toxin was even more than 4 mg/kg. Few examples are listed as follow:
- Muto, Y.; Tani, Y.; Okada, T., et al. Effects of T-2 toxin treatment on CCl4 induced hepatic necrosis in mice. Toxicol. Pathol. 2002, 15, 161-166.
- Chaudhari, M.; Jayaraj, R.; Santhosh, S.R.; et al. Oxidative damage and gene expression profile of antioxidant enzymes after T‐2 toxin exposure in mice. Biochem. Mol. Toxicol. 2009, 23, 212-221.
- Wu, J.; Yang, C.; Liu, J.; et al. Betulinic acid attenuates T-2-toxin-induced testis oxidative damage through regulation of the JAK2/STAT3 signaling pathway in mice. Biomolecules 2019, 9, 787.
- Chaudhari, M.; Jayaraj, R.; Santhosh, S.R.; et al. Oxidative damage and gene expression profile of antioxidant enzymes after T‐2 toxin exposure in mice. Biochem. Mol. Toxicol. 2010, 23, 212-221.
- Yang, J.Y.; Zhang, Y.F.; Liang, A.M.; et al. Toxic effects of T-2 toxin on reproductive system in male mice. Toxicol. Ind. Health 2010, 26, 25-31.
Point 3: By my opinion, better to show the complete error bars on the figures instead of the only upper half.
Response 3: We changed the figures with the complete error bars.
Reviewer 3 Report
- The list of abbreviation used in the text added at the beginning of the article would increase readability of the manuscript.
- Introduction: Addition of table showing changes caused by T2 toxin with appropriate references would increase the readability of the manuscript
- Please add the number and date of the receipt of agreement from ethical committee
- Why the Authors used BA in doses 25 mg/kg, 0.5 mg/kg and 1 mg/kg, and not for example 2 mg/kg, 2.5 mg/kg or 3 mg/kg Please justify the using of such doses . If they result from previous studies, it should be mentioned. The same comment concerns the dose of T2 toxin.
- T2 toxin gets into the living organisms mainly with food. Why the Authors decided to give T2 toxin through an intraperitoneal injection?
- Catalogue numbers of reagents (especially antibodies and ELISA kits) should be added
- Figure 4 Microphotographs (especially changes marked with arrows) are not readable. Maybe photographs with higher magnification would solve the problem?
- Clear conclusion and summarization of obtained results should be added at the end of discussion.
In spite of the mentioned above suggestions, the reviewer thinks that the manuscript after correction (minor revisions) may be published.
Author Response
Responses to Reviewer 3 Comments
Point 1: The list of abbreviation used in the text added at the beginning of the article would increase readability of the manuscript.
Response 1: Thanks for the comments. We added the list of abbreviation.
Point 2: Introduction: Addition of table showing changes caused by T-2 toxin with appropriate references would increase the readability of the manuscript
Response 2: Thanks for the comments. Change was made accordingly in line 47.
Point 3: Please add the number and date of the receipt of agreement from ethical committee.
Response 3: Thank you for your reminder. The date and number of the receipt of agreement from ethical committee have been added which can be foud in “4.2 Experimental design”.
The project approval code is 43321820, approval date is 1st March 2018.
Point 4: Why the Authors used BA in doses 0.25 mg/kg, 0.5 mg/kg and 1 mg/kg, and not for example 2 mg/kg, 2.5 mg/kg or 3 mg/kg. Please justify the using of such doses. If they result from previous studies, it should be mentioned. The same comment concerns the dose of T-2 toxin.
Response 4: The doses of BA and T-2 toxin were determined based on previous studies and preliminary experiments. We have added relevant references in the manuscript in line 310-311.
Point 5: T-2 toxin gets into the living organisms mainly with food. Why the Authors decided to give T2 toxin through an intraperitoneal injection?
Response 5: Thanks for the comments. Oral gavage may be closer to simulating the damage of T-2 toxin intake to animals. Of course, this question raised by the reviewer is pretty good, and it is still very helpful for our future research. In our study, we choose to intraperitoneally injected with T-2 toxin was based on our previous study, and the previous study also have good experimental results. Besides, modeling by intraperitoneal injection of T-2 toxin is actually also quite commonly used on other research work published in well-known scientific journals. Few examples are listed as follow:
- Zhang, Y.F.; Yang, J.Y.; Meng, X.P.; et al. l-arginine protects against T-2 toxin-induced male reproductive impairments in mice. Theriogenology. 2019, 126, 249-253.
- Ahmadi, A.; Poursasan, N.; Amani, J.; et al. Adverse effect of T-2 toxin and the protective role of selenium and vitamin E on peripheral blood B lymphocytes. J. Immunol. 2015, 12, 64-69.
- Yang, J.Y.; Zhang, Y.F.; Liang, A.M.; et al. Toxic effects of T-2 toxin on reproductive system in male mice. Ind. Health. 2010, 26, 25-31.
- Chaudhari, M.; Jayaraj, R.; Santhosh, S.R.; et al. Oxidative damage and gene expression profile of antioxidant enzymes after T‐2 toxin exposure in mice. Biochem. Mol. Toxicol. 2010, 23, 212-221.
- Li, M.; Harkema, J.R.; Islam, Z.; et al. T-2 toxin impairs murine immune response to respiratory reovirus and exacerbates viral bronchiolitis. Appl. Pharmacol. 2006, 217, 76-85.
Point 6: Catalogue numbers of reagents (especially antibodies and ELISA kits) should be added.
Response 6: Change was made accordingly.
Point 7: Figure 4 Microphotographs (especially changes marked with arrows) are not readable. Maybe photographs with higher magnification would solve the problem?
Response 7: Thanks for your comments. We re-made the picture of H&E, the picture was enlarged, and the lesion area can be seen more clearly.
Point 8: Clear conclusion and summarization of obtained results should be added at the end of discussion.
Response 8: Thanks for the comments. Change was made accordingly.
Round 2
Reviewer 1 Report
The manuscript has been improved remarkedly, the open questions concerning version 1 are appropriately answered. Especially the morphological figures are now clear and focus the observations of the authors.
Two small aspects: It is indicated that the morphological pictures are taken at a magnification of 40x. Obiously the object lens was 40x. The overall magnification was much higher - and in comparison to version 1 the pictures now are even magnified to a higher extent - as seen by the scale bar for 50 micrometers. Thus - either give the correct total magnification (object lens, optic lenses to adapt the camera, camera picture in mm and final size as it is shown in the manuscript) - or drop the magnification and relay on the scale bar.
And - give short note how many of these pictures were screened - if it was semi-quanitative indicate this.
Overall there are many other improvements
Author Response
Responses to the comments from reviewer 1
Responses to Reviewer 1 Comments
Point 1: Two small aspects: It is indicated that the morphological pictures are taken at a magnification of 40x. Obiously the object lens was 40x. The overall magnification was much higher - and in comparison to version 1 the pictures now are even magnified to a higher extent - as seen by the scale bar for 50 micrometers. Thus - either give the correct total magnification (object lens, optic lenses to adapt the camera, camera picture in mm and final size as it is shown in the manuscript) - or drop the magnification and relay on the scale bar.
Response 1: First of all, thank you very much for your suggestions, and this was very earnest and responsibility. According to your suggestion, we made the corresponding modification. We used 50 µm scale to indicate the size of the pictures, and deleted the “(H&E staining, 40× magnification)” in line 128.
Point 2: And - give short note how many of these pictures were screened - if it was semi-quantitative indicate this.
Response 2: Thanks for the comments. We did not use semi-quantitative to analysis, and we mainly to describe the lesions of the slices through microscope observation.
